# Pollution and Potential Ecological Risk Evaluation of Heavy Metals in the Bottom Sediments: A Case Study of Eutrophic Bukwałd Lake Located in an Agricultural Catchment

**DOI:** 10.3390/ijerph20032387

**Published:** 2023-01-29

**Authors:** Marcin Sidoruk

**Affiliations:** Department of Water Resources and Climatology, University of Warmia and Mazury in Olsztyn, Plac Łódzki 2, 10-719 Olsztyn, Poland; marcin.sidoruk@uwm.edu.pl; Tel.: +48-895-234-351

**Keywords:** lake sediment, heavy metals, potential ecological risk

## Abstract

Metals are natural components of the earth’s crust and are essential elements in the metabolism of fauna and flora. However, some metals at high concentrations may pose an ecological risk. Ecological risk analysis is one of the best methods for detecting potential metal pollution problems and its impact on ecosystems. This study analyzed the potential ecological risk and contamination from heavy metals (Cd, Cu, Pb, Ni, Cr, and Zn) in bottom sediments. It analyzed the spatial variability of heavy metal concentrations in the bottom sediments of Lake Bukwałd. The potential ecological risk index (RI) was used to assess the impact of pollutants accumulated in bottom sediments on the environment. In addition to RI, the geochemical index (Igeo) and contamination factor (CF) were also calculated. The pollutant loading index (PLI) was used to compare the average content of metals in the bottom sediments of lakes. The obtained results indicate that agricultural activity determined the quality of the bottom sediments of the reservoir and the spatial content of trace metals in them. Higher concentrations of elements were found in sediments collected from the agricultural catchment, whereas the lowest concentrations were observed near the outflow from the lake. The calculated RI value was 153.3, representing a moderate ecological threat risk. The concentration of cadmium had the greatest influence on the value of the indicator. The analysis of the scale of pollution of bottom sediments using the geochemical index showed that the bottom sediments in terms of the content of most of the trace metals tested are class II. In Cu and Zn, the Igeo index classified these deposits as class 0 and class I. The highest CF values were determined for Cr, Ni, and Pb and classified as significant contamination. The remaining elements were classified as moderately contaminated. The designated PLI was 2.49, suggesting immediate action to reduce pollution and prevent the degradation of the lake.

## 1. Introduction

Climate is the most important natural factor influencing the development of sediment deposition processes in lakes. The functioning of the lake ecosystem itself, the use of the lake basin, and the functioning of terrestrial ecosystems developing in the lake basin depend on it. For this reason, a completely different way of functioning of lake ecosystems is observed in areas with a humid tropical climate, different in sub-tropical climate or in lakes of the temperate zone. The differences in the functioning of lake ecosystems in individual zones are so large that they affect the development of sedimentation processes and thus are reflected in the different lithologies of lake sediments [1,2,3].

Various local and regional factors have a major influence on the chemical composition of lake bottom sediments. Among the most important are the geologic structure and lithology of the lake’s drainage basin, which decisively influence the chemical composition of the inflowing surface and ground waters, as well as the nature of the drainage basin’s use [4,5,6]. The type of soils found in the watershed and how they are used significantly influence the intensity of denudation and determine the sedimentation rate [7,8,9]. Vegetation cover in the lake catchment not only causes changes in the intensity of mechanical and chemical denudation but also affects changes in the mutual relationships of these two types of denudations. As a result, it affects the inflow of various dissolved and suspended substances into the lake [7,10].

Heavy metal pollutants are considered a significant threat to aquatic ecosystems due to their long-term effects, longevity, ability to enter the food chain, and toxicity [11,12]. Once in the water system, trace metals can settle in suspended solids and bottom sediments, which can be a long-term source of pollution [13]. They can also have negative impacts on aquatic ecosystems, the food chain, and human health [14]. Numerous studies indicate that only a portion of the trace metals remains in the aquatic environment, while the majority accumulates in the bottom sediments [15]. Trace metals bind to bottom sediments in the hyporheic zone through a number of processes, including co-precipitation, adsorption on the particle surface, ion exchange, hydrolysis, and sedimentation processes in organic material. The sources of environmental pollution with trace metals are industrial emissions, traffic emissions, municipal management, and agriculture (use of phosphate fertilizers, lime, and pesticides) [16,17].

The accumulation of trace metals in bottom sediments is a constant threat to the elements of the natural environment. Trace metals released from bottom sediments can pose a threat to plant and animal organisms in water, and in the case of the use of water contaminated with trace metals for irrigation, also to agricultural crops. It has been proven that some metals that accumulate in bottom sediments are hazardous, even in small quantities. For this reason, it is extremely important to analyze the indicators describing soil sediments’ quality. Thanks to these indicators, it is possible to determine the degree of pollution risk as well as the degree of toxicity of heavy metals in the aquatic ecosystem [6,9,18].

The aim of the study was to analyze (a) the assessment of the potential ecological risk and contamination with heavy metals (Cd, Cu, Pb, Ni, Cr, and Zn) in the bottom sediments and (b) the spatial variability of heavy metal concentrations in the bottom sediments of Bukwałd Lake.

## 2. Materials and Methods

Lake Bukwałd (53°52′23.6″ N; 20°20′54.3″ E) in the north of Poland in Central Europe in the area of the Olsztyn Lake District was selected for the study of the contamination of bottom sediments with heavy metals and the assessment of the potential ecological risk. The area of the lake district is characterized by a varied young-glacial relief connected with the Łyna lobe and located in the area of the last Baltic glaciation. The relief in the study area was formed after the Pomeranian phase of the Wüerm glaciation (Pleistocene), when the water of the glacier melted. The area is largely covered by glacial deposits, including clays and silts (in the uplands) and fluvioglacial sands and gravels (above the studied lake). In the immediate catchment area of Lake Bukwałd, the surface layer consists of light silty loam, which changes to medium loam, light loam, or light clay at a depth of about 0.5 m. Near the outflow, light loamy sands change into light loam.

As for climatic conditions, the studied lake is located in a cool and humid climate with an average annual temperature of 6.5–7.3 °C. An important feature of the climate is its transitional character between the maritime and continental climates. The average annual precipitation is 605 mm, with variations from 450 mm to over 700 mm [19,20].

The studied lake has four tributaries and is a reservoir with an area of 36.2 ha, a maximum depth of 12.4 m, and an average depth of 5.4 m (Table 1). In the studied lake, the depth index was 0.44, proving that the lake shell has a concave parabolic shape, and its bottom is relatively flat, with great depths occurring on a small reservoir area.

The catchment area of the lake is a hilly area with a difference between the highest and lowest point in the catchment area of 23.5 m. The whole catchment area of the lake, which is 28.98 km^2^, was divided into four parts, which differ in area and use (Figure 1A). The agricultural catchment, with a total area of 13.33 km^2^, is located in the northwestern part of the lake catchment. The area is drained by two watercourses. The first is 4.95 km long, and the second is 1.75 km long. The longer tributary flows through a village inhabited by about 150 people. Another sub-basin with an area of 0.69 km^2^ consists of wetlands forming a low peat bog. The area of forest areas in the whole catchment of Bukwałd Lake is 9.56 km^2^. The remaining part of the catchment consists of fallow and sparsely urbanized areas (rural development).

On the northern side of the lake, these areas are 4.43 km^2^, while on the southern side, they are 0.97 km^2^. In the northern part of the basin, light clays transitioning to medium clays predominate, while in the southern part, light and heavy clay sands transitioning to light clays and light clay sands predominate. In the arable land, canola and wheat are grown in layers.

Samples of bottom sediments from Lake Bukwałd for physical and chemical analyses were collected from 5 cross transects in which 3 sediment collection stations were designated (Figure 1B). The selection of points was based on a bathymetric plan and was planned to obtain as complete a picture as possible of the diversification of sediments and the rate and nature of their accumulation as a function of the shape of the bottom and the shape of the lake basin, tributaries, and the use of its sub-basins. Surface samples of bottom sediments were collected from a depth of 0–10 cm using an Ekman trap (samples with an area of 250 cm^2^ were collected). Due to the strong hydration of the sediments, which resulted in a small amount of mud after drying, it was decided to take three samples from each site, which were then averaged into one. After the samples were dried at room temperature, they were crushed, and then their particle size was measured by wet laser diffraction using a Mastersizer 3000 particle analyzer. Dried sediment samples were mineralized in a Multivave 3000 microwave oven (Anton Paar, Austria). A total of 0.5 g of the sample was weighed into a fluorinated polymer vessel, and 5 mL of 65% HNO_3_ and 15 mL of 35–38% HCl were added. The vessels were covered with a shield and then placed in the rotor. Mineralization took about 110 min and was carried out at a maximum temperature of 240 °C, a maximum pressure of 60 bar, and a maximum power of 1400 W. After mineralization, the samples were transferred to 100 mL volumetric flasks and filled with deionized water.

The organic matter content was quantified on the basis of mass loss after ignition at 550 °C and was added to the Materials and Methods. The amount of carbon was determined on the basis of the organic matter content with an average factor of 1.724, as described by Pribyl [21].

The content of elements (Cd, Cu, Pb, Ni, Cr, and Zn) was determined by induced plasma atomic emission spectrometry (ICP-OES iCAP- PRO, Thermo Fisher Scientific, Waltham, MA, USA). A multi-element standard solution with a concentration of 100 mg/L of each element was used to generate the element calibration curves.

To determine the total and partial catchment areas of Lake Bukwałd, the SCALGO Live program from SCALGO ApS Denmark was used. In order to determine the spatial distribution of the analyzed metals in the bottom sediments of the lake, IDW interpolation of the obtained results was performed using QGIS 3.22 software, which generated an inverse distance weighted (IDW) interpolation of a point vector layer. The sample points are weighted in the interpolation such that the influence of one point relative to another decreases with distance from the unknown, determined point. In order to identify the environmental factors influencing the content of trace metals in the bottom sediments of the studied lake, principal component analysis (PCA) was used, i.e., multivariate statistical analysis using the software Canoco 5.0 for this purpose. The significance of the differences in density and organic and mineral matter content in the bottom sediments between the different parts of the lake was assessed using Tukey’s non-parametric test of variance (*p* ≤ 0.05).

The potential ecological risk (*RI*) index proposed by Hakanson [22] was used to evaluate the environmental impact of contaminants accumulated in bottom sediments. The value *RI* was calculated using the following formulas:(1)$$C_{f}^{}=\frac{C_{surface}^{}}{C_{reference}^{}}$$


*E_r_* = *T_r_* · *C_f_*
(2)



(3)$$RI=\overset{n}{\underset{i=1}{\sum }}E_{r}^{}$$
where *C_surface_* is the mean content of the heavy metal (mg·kg^−1^); *C_reference_* is the preindustrial reference value for the substance (mg·kg^−1^); *E_r_* is the potential risk of individual heavy metal; *T_r_* is the toxic-response factor for a given heavy metal (Cd = 30, Cu = Pb = Ni = 5, Cr = 2, and Zn = 1); and *C_f_* is the contamination coefficient (mg·kg^−1^).

The results achieved by Er and RI were divided into classes according to the range shown in Table 2.

In addition to the potential ecological risk (RI) index, the geochemical index (*Igeo*) of the sediment samples was also calculated based on the following formula of Muller [23]:(4)$$Igeo=log2\left(\frac{C_{m}}{1,5GB}\right)$$
where *Igeo* is the geochemical index; *C_m_* is the concentration of the analyzed metal (mg·kg^−1^); and *GB* is the geochemical background (mg·kg^−1^) [21].

The contamination factor (*CF*) was calculated with the use of Muller’s formula [23]:(5)$$CF=\frac{Cm}{GB}$$
where *Cm* is the concentration of the analyzed metal (mg·kg^−1^) and *GB* is the geochemical background (mg·kg^−1^) (Cd = 0.5, Cu = 6, Pb = 10, Ni = 5, Cr = 5, and Zn = 48) [21].

The values proposed by Bojakiewicz and Sokołowska [24] were adopted as the geochemical background value. PLI is an indicator used to compare the average content of metals in the bottom sediments of lakes. The pollutant load index (PLI) was calculated by the following formula [25,26,27]:(6)$$\mathrm{PLI}=\sqrt[{6}]{{\mathrm{CF}}_{Cu}\;\cdot {\;\mathrm{CF}}_{Ni}\;\cdot {\;\mathrm{CF}}_{Cd}\;\cdot {\;\mathrm{CF}}_{Cr}\;\cdot {\;\mathrm{CF}}_{Pb}\;\cdot {\;\mathrm{CF}}_{Zn}}$$

The results achieved by Igeo, CF, and PLI were divided into classes according to the range shown in Table 3.

## 3. Results

The granulometric composition of the collected bottom sediments was similar at all sites (Table 4). The content of the finest fraction (clay—<2 µm and silt—2–50 µm) ranged from 0.1 to 0.3% for clay and 2.4 to 9.7% for silt. In the coastal areas of the lake, the proportion of these fractions was slightly higher than in the central part of the reservoir. The dominant fraction of bottom sediments, accounting for more than 50% of the total fraction at most monitoring points, was fine sand (100–250 µm). Only at points B3-1, B3-2, and B5-3 was its fraction lower and amounted to about 30%. The medium sand fraction (250–500 µm) dominated at these measurement points, ranging from 58.5 to 62.4%. The examined sediments did not contain any fractions above 500 µm.

The bottom sediments of Lake Bukwałd were characterized by high hydration and low consolidation. Their hydration ranged from 66.25 to 86.82%, with lower hydration of sediments observed on the northern side of the lake than on the southern side (Table 5). In this part of the reservoir, there were also sediments with higher mineral matter content. The organic matter content ranged from 8.1 to 19.9%. The highest content of organic matter was observed in the sediments from the deep pit and the lowest at the outlet from the north side of the lake. There were no statistically significant differences between the measurement points in terms of bulk density and sediment hydration. Statistically significant differences were found in the mineral and organic matter content of the sediments collected on the west side of the lake, while the bottom sediments at the other points showed no statistically significant differences.

The content of trace metals in the sediments of Bukwałd Lake is presented in Table 6. The concentration of metals in the examined bottom sediments of Lake Bukwałd can be arranged in the following order from top to bottom: Zn > Pb > Ni > Cr > Cu > Cd. 

The calculated coefficient of variation was low (12–31%). Most of the samples studied were below 25%, which means very low variability in the results obtained. Only in the case of Cr was it 31%. This means that the obtained results of spatial chromium concentration in sediments were characterized by medium variability.

The pH of the bottom sediments of Lake Bukwałd was neutral and their pH was at a relatively uniform level and did not exceed 7.40 (Table 7). However, considerable fluctuations were observed in the nitrogen concentrations. Its content in the sediments ranged from 0.30 to 9.30 mg·kg^−1^. Higher nitrogen concentrations were found in the sediments from the agricultural catchment and the lowest in the sediments collected near the outflow of the lake. The calculated coefficient of variation was high (88%), which means that the obtained results of spatial nitrogen concentration in sediments were characterized by high variability. Such values may indicate an anthropogenic source of nitrogen enrichment in the sediments. 

The multidimensional principal component analysis (PCA) technique was used to obtain a synthetic picture of the relationship between a large number of variables. This analysis showed that the first two factors accounted for 91.62% of the cumulative variance of the species–environment relationship, and the other factors did not contribute significantly to the information contained in the data matrix (Figure 2).

Based on principal component analysis (PCA), a positive correlation was found between the content of Zn, Ni, and Cr in surface sediments and the organic matter content and hydration of bottom sediments. The correlations of these metals may indicate their similar sources. In turn, a negative correlation of cadmium, lead, and copper with the bulk density of soil sediments was found.

The spatial concentrations of Cd and Cu in the bottom sediments of Lake Bukwałd showed little variation (Figure 3). The analysis of the content of these elements in the bottom sediments of Bukwałd Lake allowed us to distinguish two local maxima. Both are located on the agricultural catchment side, i.e., in the bay in the northern part of the reservoir (2.02 mgCd·kg^−1^, 10.25 mgCu·kg^−1^) and on the northeastern side of the lake near the longer agricultural inflow (1.95 mgCd·kg^−1^, 9.60 mgCu·kg^−1^). The lowest cadmium content in the sediments was found on the northwest side of the lake (1.33 mgCd·kg^−1^) and copper on the southwest side of the lake at the outlet from the reservoir (5.00 mgCu·kg^−1^).

In the case of Ni, Cr, and Zn, the spatial distribution was similar in the bottom sediments of the lake. Higher concentrations occurred in the sediments collected in the northeastern part of the lake on the agricultural catchment side, and their maxima occurred near the watercourses draining this part of the catchment (25.45 mgNi·kg^−1^, 29.55 mgCr·kg^−1^, and 112.92 mgZn·kg^−1^). The lowest content of these elements was found in sediments collected on the west side of the lake near the outflow (14.45 mgNi·kg^−1^, 8.20 mgCr·kg^−1^, and 45.91 mgZn·kg^−1^). For lead, the maximum was observed in sediments near the longer watercourse draining the agricultural catchment (56.00 mgPb·kg^−1^), while the lowest values were found at the outflow of the lake (24.00 mgPb·kg^−1^). 

The results of the evaluation of the potential ecological risk factor (Er) and the potential ecological risk index (RI) are presented in Table 8. The obtained data show that the potential ecological risk index (Er) for Pb, Zn, Cr, Ni, and Cu was low (Er < 40) and represented a low risk. Only in the case of Cd, the value of Er index was higher (98.41), representing a considerable risk (Table 7). The calculated ecological risk index can be arranged in the following order: Cd > Ni > Pb > Cr > Cu > Zn. The calculated potential ecological risk index (RI) for the trace metals accumulated in the soil sediments was 153.3, representing a moderate ecological threat risk. Cadmium (65%) had the greatest influence on the value of the index, and zinc had the least influence.

The calculated geoaccumulation indices showed insignificant variability, ranging from −0.28 to 1.53 (Table 9). The lowest value was found for Cu (class 0), indicating that the bottom sediments of the lake are not contaminated with this element. A low value of the indicator was also found for Zn (0.34), indicating a class I classification for unpolluted to moderately polluted sediments. The Igeo index value for the other elements placed them in class II as moderately contaminated sediments.

The highest values of the contamination factor (CF) were determined for Cr, Ni, and Pb, and according to this indicator, the studied sediments were classified as considerable contamination (class III) (Table 10). The other elements were classified as moderate contamination (class II). The determined pollutant loading index (PLI) in the sediments of Bukwałd Lake was 2.49, suggesting that immediate measures should be taken to reduce pollution and prevent the degradation of the basin.

Comparing the Igeo index with the CF index, we find that the bottom sediments of Cu in the lake were assigned to class 0 as uncontaminated sediments according to the Igeo index and to class II, i.e., moderately contaminated, according to the contamination factor (CF). In the case of the other elements, both indices evaluated the contamination of the bottom sediments of the lake in a similar way.

## 4. Discussion

Surface sediments are an important element of lake ecosystems. In shallow water systems, such as lakes and reservoirs, the intensive exchange of material between soil and water can determine the nutrient flux and productivity of the entire ecosystem [28,29]. Granulometric composition is an important characteristic of bottom sediments because the ability of bottom sediments to concentrate and retain trace elements depends on it [9,30,31]. The tests carried out showed that in the analyzed bottom sediments, the content of the smallest fractions was low, averaging 0.2% for clay and 4.9% for silt. In the coastal zones of the lake, the content of these fractions was slightly higher than in the central part of the reservoir. The bottom sediments of the lake were dominated by the fine sand fraction and, at some measurement points, by the medium sand. It was found that the bottom sediments of Bukwałd Lake are characterized by strong hydration and low consolidation. On the northern side of the lake (on the side of the agricultural catchment), lower hydration of sediments was observed than on the southern side. This part of the reservoir also contained sediments with a higher content of mineral matter. The higher content of mineral matter in the bottom sediments of the north side of the reservoir was caused by the presence of arable land on this side of the reservoir and, thus, by easier leaching of the soil from the substrate as a result of the erosion process. The substantial land drainage occurring in this part of the watershed further facilitates this process. A similar relationship has also been noted by other researchers [32,33,34]. 

Due to the low depth index of the studied lake (0.44), the bottom sediments are easily broken up and float in the water, affecting their hydration and bulk density. Significant hydration of bottom sediments affects the processes of re-suspension and exchange of sediment–water components in the surface zone of bottom sediments. Many authors [35,36,37] show that sediments from the deep usually accumulate more organic matter than in the zones of shallower lakes, which was also confirmed by the research conducted.

The major natural source of trace metals in lake bottom sediments is weathering and erosion of rocks and soils, and the amount of metals carried by this source determines the content in a given area, which is considered the local geochemical background. The estimation of the natural metal content in lake sediments for a given region is very important because it is the reference value for determining the degree of contamination of the sediments. A large amount of particulate matter is transported into lakes by watercourses. Streams flowing into lakes bring pollutants in both dissolved and suspended forms. The percentage of dissolved forms in the total transport of pollutants expresses the so-called transport index, which depends on the amount of wastewater discharged into the rivers and the size of the river. The differences in the content of individual trace metals in the studied bottom sediments are undoubtedly influenced by different hydrodynamic conditions of water flow in the reservoir, as well as by the intensity of runoff from the immediate catchment area of the reservoir (the southern part of the catchment area is predominantly farmland). According to many authors [8,38,39,40], phosphorus and compound fertilizers used in agriculture can be a significant source of trace metals (Cd, Cu, Ni, Pb, and Zn) in soil sediments in agricultural and peri-urban areas. Such a relationship was observed in the case of the studied Bukwałd Lake, where higher concentrations of trace metals were observed in sediments collected on the side of the catchment used as farmland than on the side of the catchment used in other ways. Metals present in the surface layer of bottom sediments can be reactivated under unfavorable conditions and become a source of secondary pollution [9,32,40]. Metals dissolved in water are biologically absorbed and accumulated by living organisms. Some of these elements are excreted along with metabolites, forming soluble forms. After the death of the organisms, the dead substances enter the soil sediments in the form of organic tryptone. In the soil zone, where anaerobic conditions often prevail, re-reduction and conversion to soluble forms may occur [10,41,42].

The tests carried out showed a negative correlation between Cu, Cd, Pb, and the bulk density of the sediments, and no correlation was found with the organic matter. Similar results were obtained by Frémion et al. [43], who also found no correlation between trace metals and the organic content of the bottom sediments. The concentrations of Cr, Ni, and Zn were found to be strongly correlated with the organic matter and hydration of the soil sediments. It was also found that the concentrations of these elements were negatively correlated with the content of silt and clay. Sojka et al. [44] came to different conclusions and found stronger correlations between trace metals and the content of silt and clay. Farhat and Aly [45] suggest that organic matter is more important than grain size for the distribution of trace metals in soil sediments.

The analysis of the principal components of the PCA showed that the elements studied formed two separate groups in which a strong correlation was found between them. The first group consisted of Cr, Ni, and Zn, and the second of Cu, Cd, and Pb. Sojka et al. [44] and Wang et al. [46] indicate that trace metals share a common geochemical behavior and originate from similar pollution sources when the correlation coefficient between them is higher. The lack of correlation between them indicates that the content of these metals is not controlled by one factor but by several. This means that the content of trace metals in the bottom sediments of Lake Bukwałd was influenced by several factors, e.g., granulometry, organic matter content, sediment hydration, or fertilizers and pesticides used. In addition, the investigations revealed a clear influence of the areas directly adjacent to the reservoir on the content of trace metals in the bottom sediments of the lake.

The content of trace metals in soil sediments is associated with the possibility of ecological threats. The ecological impacts of the contamination of surface sediments with toxic metals are manifold. Some metals (Zn, Cu, and Ni) are essential for biological systems. However, they can also be toxic if critical concentrations are exceeded [47,48]. Cadmium, lead, and chromium have no practical biological significance when present in sediments in natural amounts, but in high concentrations, they are toxic to organisms [49]. The studies showed that the potential ecological risk index (Er) for most metals was low and represented a low risk. Only in the case of Cd was the value of the Er index higher and classified as a significant risk. The value of the potential ecological risk (RI) index, representing moderate ecological risk, was influenced by the cadmium content in the soil sediments. The increased cadmium content in the bottom sediments of Lake Bukwałd is mainly related to anthropogenic pollution originating from the agricultural catchment area. Cadmium enters surface waters in agricultural areas together with fertilizers and pesticides used in agriculture and then accumulates in bottom sediments [50,51]. It should be emphasized that Cd (in excessive concentrations) is highly toxic, especially to humans and animals (and less toxic to plants), while Zn and Cu are the opposite [12,46,52].

To assess the degree of contamination of bottom sediments with metals, many authors use geochemical criteria in relation to the geochemical background, i.e., the content of elements present in the sediments under natural conditions (geochemical index and contamination factor) [23,38,53]. The obtained values of the enrichment coefficient Igeo and CF for selected trace metals accumulated in the bottom sediments of the studied lake indicate natural (weathering of the rock material) and anthropogenic (inflow of pollutants from the agricultural catchment) sources of the identified elements [41,52,54]. In the bottom sediments of Lake Bukwałd, the Igeo and CF index values were low and classified as unpolluted and moderately polluted sediments. Similar index values in water reservoirs fed from agricultural, forested, or sparsely urbanized catchments were also found in the bottom sediments of lakes located under similar geographical and climatic conditions. However, despite the low values of these indices, the pollutant loading index (PLI) determined in the sediments of Lake Bukwałd was 2.49. This suggests that immediate measures should be taken to reduce pollution and prevent the degradation of the catchment. A comparison of the Igeo indices calculated for the sediments of Lake Bukwałd with the indices calculated for lake sediments in urbanized catchments showed that their values are mostly higher (Table 11). Only for Cr and Ni were the index values lower than in the studied lake. A similar relationship was found for the contamination factor (CF) [49,55,56,57,58,59].

In order to limit the deterioration of Lake Bukwałd, a number of measures should be taken. An important element of improvement would be the introduction of preventive measures in the catchment area. Preventive principles include adherence to the principles of good agricultural practice, i.e., precise fertilization and maintenance of the soil in a high culture that meets the needs of the plants. Agricultural activities should be supported by appropriate meliorations, such as drainage water reservoirs or ditches around the lake. An important direction of activities is also the appropriate design of landscape structure, erosion control methods in the development and use of slopes, and the establishment of biogeochemical barriers (e.g., the introduction of the biological development of coastal zones).

## 5. Conclusions

The spatial concentration of trace metals in the bottom sediments of Lake Bukwałd was characterized by small differences. The results obtained indicate that agricultural activity has a decisive influence on the quality of the bottom sediments of the reservoir and the spatial content of trace metals in them. The use of the lake basin was found to have a decisive influence on the concentration of trace elements in the bottom sediments of the lake. The highest concentrations of trace metals were found in sediments taken from the side of the catchment area used for agriculture. The bottom sediments from the other sub-catchments (forest areas, wetlands, fallow land, and sparsely populated areas) had lower levels of the metals studied. The calculated potential ecological risk index (Er) was low for most elements and represented a low risk. Only in the case of Cd was the value of the Er index higher and represented a significant risk. The calculated potential ecological risk index (RI) for the trace metals accumulated in the bottom sediments was 153.3 and represented a moderate risk of ecological threat. The cadmium concentration was the decisive factor for the value of the indicator. Analysis of the extent of contamination of the bottom sediments using the geochemical index (Igeo) showed that the bottom sediments from Lake Bukwałd belong to class II in terms of the content of most of the trace metals tested. Only in the case of Cu and Zn did the calculated Igeo index classify these sediments into class 0 (uncontaminated sediments) and class I (uncontaminated to moderately contaminated sediment). The highest contamination factor values (CF) were determined for Cr, Ni, and Pb and classified as significant contamination (class III). The remaining elements were classified as moderate contamination (class II). The contaminant load index (PLI) in the sediments of Lake Bukwałd was 2.49. This suggests that immediate action should be taken to reduce pollution and counteract the degradation of the basin.

## Figures and Tables

**Figure 1 ijerph-20-02387-f001:**
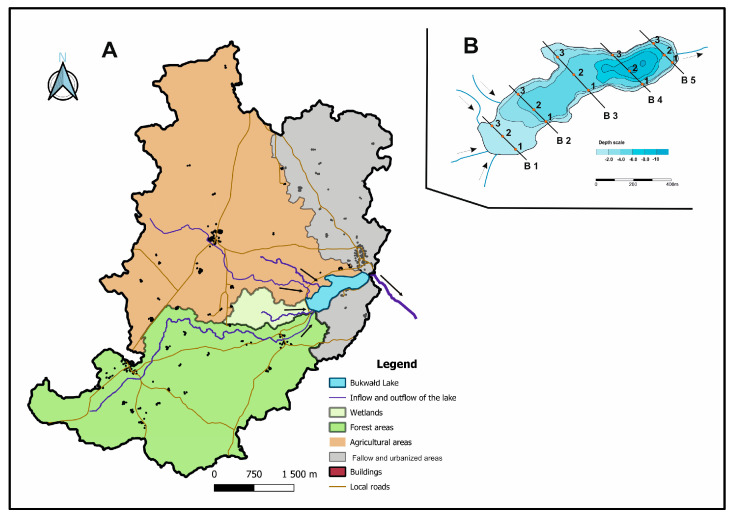
Land use in the catchment area of Lake Bukwałd (**A**). The location of sediment sampling sites is shown on the bathymetric map of Lake Bukwałd (**B**).

**Figure 2 ijerph-20-02387-f002:**
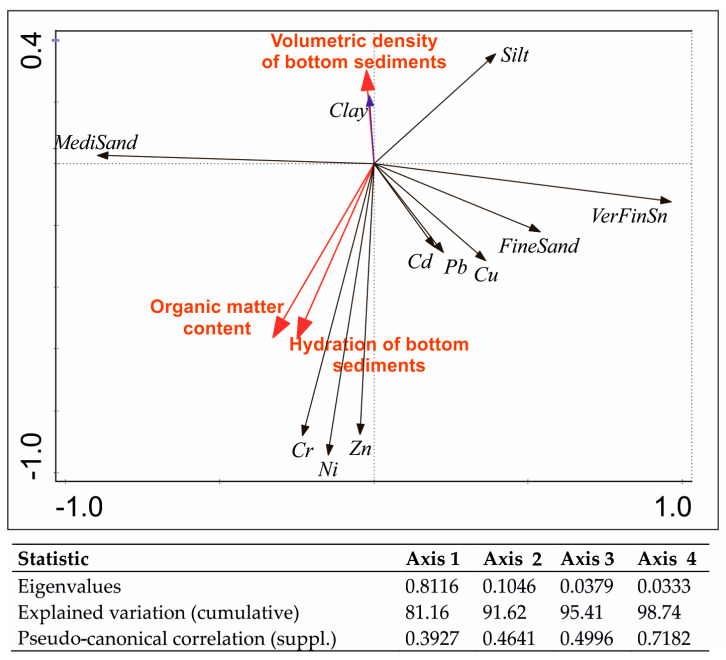
PCA order diagram to determine the influence of environmental factors on the content of trace metals in the bottom sediments of Lake Bukwałd.

**Figure 3 ijerph-20-02387-f003:**
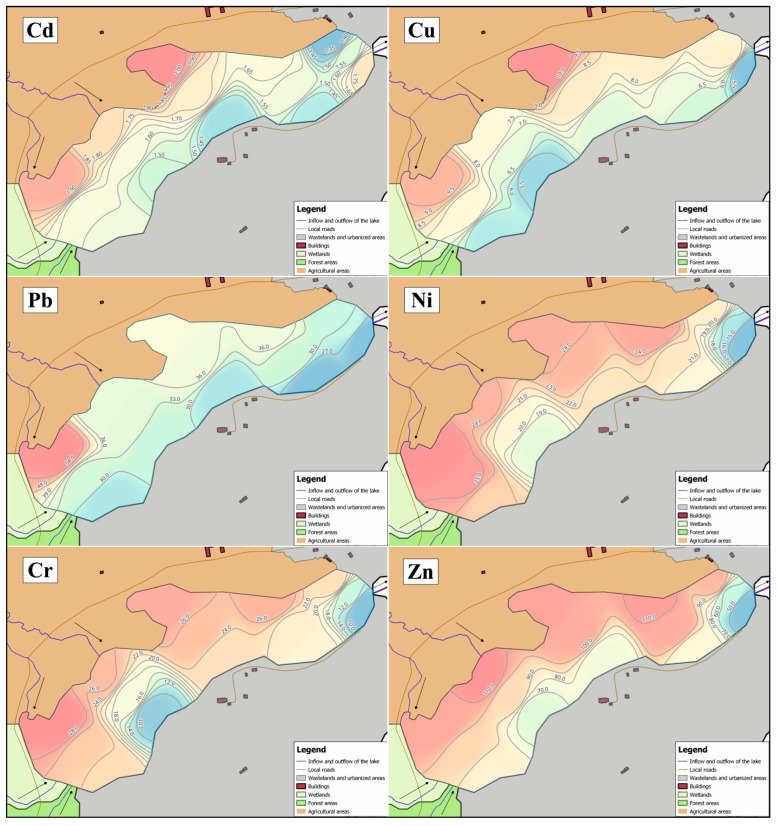
Spatial distribution of trace metals in the sediments of Lake Bukwałd (mg·kg^−1^ of dry matter).

**Table 1 ijerph-20-02387-t001:** Morphometric indicators of Bukwałd Lake.

Parameter	Value
Area [ha]	36.2
Maximum depth [m]	12.4
Average depth [m]	5.4
Relative depth	0.021
Depth indicator	0.44
Maximum length [m]	1140.0
Maximum width [m]	325.0
Shoreline length [m]	4020
Development of the coastline	1.88

**Table 2 ijerph-20-02387-t002:** Class standards for E_r_ and RI.

Potential Ecological Risk Index E_r_	Ecological Risk Level of Single-Factor Pollution	Potential Toxicity Index (RI)	General Level of Potential Ecological Risk
E_r_ < 40 40 ≤ E_r_ < 80 80 ≤ E_r_ < 160 160 ≤ E_r_ < 320 E_r_ ≥ 320	Low risk Moderate risk Considerable risk High risk Very high risk	RI < 150 150 ≤ RI < 300 300 ≤ RI < 600 RI ≥ 600	Low risk Moderate risk Considerable risk Very high risk

**Table 3 ijerph-20-02387-t003:** Range index and level of factor pollution for Igeo [25], CF [26], and PLI [27].

Geochemical Index (Igeo)		Contamination Factor (CF)		Pollutant Load Index (PLI)	
Geochemical Index Range	Level of Factor Pollution	Contamination Factor Range	Level of Factor Pollution	Pollutant Load Index Range	Level of Factor Pollution
Igeo < 0, class 0	uncontaminated sediments	CF < 1, class I	low contamination	PLI < 0.5	no drastic rectification measures are needed
0 < Igeo < 1, class I	uncontaminated to moderately contaminated sediments	1 ≤ CF ≤ 3, class I	moderate contamination		
				0.5 ≤ PLI ≤ 1	more detailed study is needed to monitor the site
1 < Igeo < 2, class II	moderately contaminated sediments	3 < CF < 6, class III	considerable contamination		
2 < Igeo < 3, class III	moderately to highly contaminated sediments	CF ≥ 6, class IV	very high contamination	PLI ≥ 1	immediate intervention to ameliorate pollution
3 < Igeo < 4, class IV	highly contaminated sediments				
4 < Igeo < 5, class V	highly to extremely contaminated sediments				
Igeo > 5, class VI	extremely contaminated sediments				

**Table 4 ijerph-20-02387-t004:** Granulometric composition of bottom sediments.

Sample No.	Percentage of Granulometric Fractions				
	<2 µm Clay	2–50 µm Silt	50–100 µm Very Fine Sand	100–250 µm Fine Sand	250–500 µm Medium Sand
B1-1	0.2	3.6	6.5	58.7	31.1
B1-2	0.1	2.7	10.6	62.5	24.1
B1-3	0.1	3.7	7.2	62.3	26.8
B2-1	0.2	3.7	9.0	58.1	29.0
B2-2	0.1	2.4	6.5	57.3	33.8
B2-3	0.3	3.6	14.5	47.3	34.4
B3-1	0.3	7.3	3.7	30.4	58.3
B3-2	0.2	3.3	4.9	29.3	62.4
B3-3	0.1	5.7	18.5	44.3	31.4
B4-1	0.2	5.2	8.0	45.2	41.4
B4-2	0.1	2.6	10.2	44.5	42.6
B4-3	0.3	8.2	13.7	46.7	31.1
B5-1	0.2	5.2	2.8	31.4	60.3
B5-2	0.3	9.72	28.2	43.3	18.5
B5-3	0.1	6.9	15.4	60.4	17.2

**Table 5 ijerph-20-02387-t005:** Density and content of organic and mineral matter in bottom sediments.

Sample No.	Bulk Density [g·cm^−3^]	Sediment Hydration [%]	Mineral Content [%]	Organic Matter Content [%]
B1-1	1.24	81.09	86.0	14.0
B1-2	1.18	82.64	84.3	15.7
B1-3	1.17	79.69	86.2	13.8
B2-1	1.11	86.28	81.7	18.3
B2-2	1.07	86.82	81.0	19.0
B2-3	1.16	68.54	89.7	10.3
B3-1	1.13	84.37	83.2	16.8
B3-2	1.17	86.05	82.6	17.4
B3-3	1.19	80.97	86.5	13.5
B4-1	1.21	76.86	88.4	11.6
B4-2	1.08	86.24	80.1 *	19.9 *
B4-3	1.39	66.25	91.6 *	8.4 *
B5-1	1.28	72.33	91.0 *	9.0 *
B5-2	1.14	80.64	86.0	14.0
B5-3	1.25	67.98	91.9 *	8.1 *

* statistically significant result, *p* ≤ 0.05.

**Table 6 ijerph-20-02387-t006:** Content of trace metals in the sediments of Lake Bukwałd (mg·kg^−1^ of dry matter).

Heavy Metal	$\bar{x}\pm SD$	Min	Max	Me	CV
Cd	1.64 ± 0.20	1.33	2.02	1.63	12
Cu	7.42 ± 1.49	5.00	10.25	7.85	20
Pb	33.03 ± 7.69	24.00	56.00	31.50	23
Ni	21.71 ± 3.36	14.45	25.45	21.80	16
Cr	21.36 ± 6.60	8.20	29.35	23.35	31
Zn	91.09 ± 21.39	45.91	112.92	96.99	24

Explanations: $\bar{x}$—arithmetic mean; SD—standard deviation; min—minimum; max—maximum; Me—median; CV—coefficient of variation [%].

**Table 7 ijerph-20-02387-t007:** Content of N, P, C, and pH in the sediments of Lake Bukwałd (g·kg^−1^ of dry matter).

Heavy Metal	$\bar{x}\pm SD$	Min	Max	Me	CV
pH	-	7.17	7.40	7.24	-
N	3.22 ± 2.83	0.30	9.30	1.73	0.88
P	0.71 ± 0.16	0.49	1.11	0.69	0.22
C	18.39 ± 2.03	15.37	22.75	17.91	0.11

Explanations: $\bar{x}$—arithmetic mean; SD—standard deviation; min—minimum; max—maximum; Me—median; CV—coefficient of variation [%].

**Table 8 ijerph-20-02387-t008:** The value of the indicator of potential ecological risk and the degree of ecological risk of pollutants of the analyzed trace metals.

Heavy Metal	Value of Potential Ecological Risk Index (E_r_)	Ecological Risk Level of Single-Factor Pollution
Cd	98.41	Considerable risk
Pb	16.58	Low risk
Zn	1.90	Low risk
Cr	8.54	Low risk
Ni	21.71	Low risk
Cu	6.19	Low risk

**Table 9 ijerph-20-02387-t009:** The geochemical index (Igeo) of sediment samples.

Heavy Metal	Value of the Geochemical Index (Igeo)	Igeo Class
Cd	1.13	class II
Pb	1.14	class II
Zn	0.34	class I
Cr	1.51	class II
Ni	1.53	class II
Cu	−0.28	class 0

**Table 10 ijerph-20-02387-t010:** The contamination factor (CF) and purity classes.

Heavy Metal	Value of the Contamination Factor (CF)	Purity Classes
Cd	1.64	class II
Pb	3.32	class III
Zn	1.90	class II
Cr	4.27	class III
Ni	4.34	class III
Cu	1.24	class II

**Table 11 ijerph-20-02387-t011:** Comparison of the indices of Igeo, PLI, and CF, which were determined for other lakes with similar and different land use.

The Method of Using the Catchment Area of Lakes	Index	Cd	Pb	Zn	Cr	Ni	Cu
Urbanized catchment area	Igeo	2.53–2.72 (2.55)	1.41–3.62 (2.45)	3.83–7.14 (4.10)	−1.3–2.71 (1.01)	0.11–2.76 (0.97)	1.10–2.5 1(1.27)
	CF	1.54–4.62 (2.12)	0.32–3.93 (1.27)	16.14–28.61 (17.20)	1.21–2.15 (1.42)	1.22–3.34 (1.56)	9.22–17.43 (10.29)
	PLI	0.69–8.95 (1.5)					
Agricultural catchment area	Igeo	0.10–0.42 (0.11)	0.11–2.34 (0.60)	0.01–2.39 (0.58)	0.02–0.32 (0.17)	0.05–2.46 (0.42)	0.11–1.66 (0.54)
	CF	0.20–4.01 (1.92)	0.26–3.56 (3.29)	0.40–2.90 (1.84)	0.10–3.20 (1.22)	0.30–16.10 (4.53)	0.80–2.79 (1.28)
	PLI	0.40–4.90 (1.35)					

Explanations: minimum–maximum (arithmetic mean).

## Data Availability

The data presented in this study are available on request from the corresponding author.

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
