# Peer review of "Pollution and Potential Ecological Risk Evaluation of Heavy Metals in the Bottom Sediments: A Case Study of Eutrophic Bukwałd Lake Located in an Agricultural Catchment"

_ijerph, 2023, doi:10.3390/ijerph20032387_

Round 1
Reviewer 1 Report
The article is devoted to the actual problem corncerning the appreciation of ecological situation in lake ecosystem (the Poland). The authors used classic methods of lake sediments sampling. The modern method of main toxic elements determination was elected. Then, three estimation indexes were calculated.
On the whole, the ecological situation in lake ecosystem was appreciated as normal. Only for two elements (Cd and Pb) in some places and sediments were obtained higher contents.
The obtained values of the enrichment coefficients (Igeo, CF) for selected heavy metals accumulated in the bottom sediments of the studied lake indicate natural (weathering of the rock material) and anthropogenic (inflow of pollutants from the agricultural catchment) sources of pollution. But, the influence of the first source of pollution may be proved more correctly. The absence of the region geological description in the article is the big lack in that relation.
The choice of elements for investigation may be proved more correctly. For example, U and Th contents in lake sediments are rather useful for appreciation of ecological situation.
1. In the article the information on geological position and source rocks of lake sediments is absent. As a matter of fact the main components of lake sediments are sand particles. And where the information about the source rocks of these particles? As to clay material: just the same. The geological information must be included.
2. 3 samples were taken from every point of sampling (line 114). It is necessary to get know the principle of their selection.
3. In the text there are repetitions of date presented in the tables (lines 189-191, the same information is in the table 6). These repetitions must be excluded.
Author Response
International Journal Of Environmental Research And Public Health
Ms. Ref. No.: ijerph-2187677
Authors: Marcin Sidoruk
Title: Pollution and Potential Ecological Risk Evaluation of Heavy Metals in the Bottom Sediments: A Case Study of Eutrophic Bukwałd Lake Located in An Agricultural Catchment
Thank you for allowing me to publish my manuscript in the International Journal of Environmental Research and Public Health and for the suggested corrections. Thank you for the reviewer's time and comments. As suggested by the reviewer, the manuscript has been changed (and hopefully improved) in many ways. The responses and changes are highlighted in red.
The answers to specific concerns are included hereafter. (Q: Question; A: Answer)
Reviewer #1
Q1) In the article the information on geological position and source rocks of lake sediments is absent. As a matter of fact the main components of lake sediments are sand particles. And where the information about the source rocks of these particles? As to clay material: just the same. The geological information must be included.
A1) First of many thanks for this comment. As suggested by the reviewer, sentences have been included in the Materials and Methods "The relief in the study area was formed after the Pomeranian phase of the Wüerm glaciation (Pleistocene), when the water of the glacier melted. The area is largely covered by glacial deposits, including clays and silts (in the uplands) and fluvioglacial sands and gravels (above the studied lake). In the immediate catchment area of Lake Bukwałd, the surface layer consists of light silty loam, which changes to medium loam, light loam or light clay at a depth of about 0.5m. Near the outflow there are light loamy sands that change into light loam." it was added.
Q2) 3 samples were taken from every point of sampling (line 114). It is necessary to get know the principle of their selection.
A2) Thanks very much for this suggestion. According to reviewer’s suggestion, in Materials and Methods at Line 114 have been added sentence “Due to the strong hydration of the sediments, which resulted in a small amount of mud after drying, it was decided to take three samples from each site, which were then averaged into one.”
Q3) In the text there are repetitions of date presented in the tables (lines 189-191, the same information is in the table 6). These repetitions must be excluded.
A3) Thanks very much for this suggestion. Duplicate information on lines 189-191 and table 6 has been removed. In line 192 the following sentence was corrected: "The concentration of metals in the examined bottom sediments of Lake Bukwałd can be arranged in the following order: from top to bottom: Zn > Pb > Ni > Cr > Cu > Cd."

Reviewer 2 Report
Please find below some comments that have to be addressed before the manuscript could be considered for publication.
Please carefully check the spelling and grammar through the whole manuscript. Some examples of typos and spelling errors: L12, L24-25, L31, L49, L65, L67, L78, L95, L99, L110, L117, L311, L405.
Abstract: The PLI is not presented in the first part of abstract, where the methodology is presented but appears in the last sentence where the results are presented. Rephrase sentence in L16-17. Rephrase “moderate risk of ecological threat”. Rephrase line 24-26.
Add “lake sediments” instead of “bottom sediments” to keywords
The authors use the term heavy metal, heavy metal pollutant, trace elements, elements. Please use a single term through the whole manuscript. I would suggest trace metals or metals.
Rephrase L34-35
L52-53 add relevant reference for the statement
L53-55 add relevant reference for the statement
L65 clarify “soil sediment” is soil or sediment?
Addition of supplementary parameters such as pH, N, P, C, major elements could give additional information on the sources of metals in the lake sediments
L88-23.5 m does not represent a considerable difference in elevation
L97 define “wasteland”
L113-115. After drying how were the sediments prepared for analysis? (Crushing? Sieving?).
The measurement method of organic matter is not presented in the Mat &Met section, yet appear in results. Introduce details on quality control approach used.
L117-sediment sample instead of soil sample
Table 2-delete Scope of
Table 3 introduce PLI levels in increasing order, similar to CF and Igeo. Revise and add references for the PLI levels.
Table 4. 1 decimal should be sufficient. The table could be presented in a more concise way or as a graph.
Table 5. Indicate for each parameter samples that have statistically different parameters (e.g. Tukey pair comparison)
L194-196 please revise or delete, as the comparable values of average and median does not show that metals have comparable concentration in the samples but that the values have a homogenous distribution.
The low (<1) eigenvalues given by the PCA indicate that this method is not appropriate for your data, or that no significant differences exist. Please revise and compare to other studies in the literature
L217-use the chemical symbol for Cd and Cu
Compare the pollution indices with those reported from other lakes with similar or different land use
Emphasize which land use have the most negative impact on the lake sediments.
Propose possible mitigation measures.
Author Response
International Journal Of Environmental Research And Public Health
Ms. Ref. No.: ijerph-2187677
Authors: Marcin Sidoruk
Title: Pollution and Potential Ecological Risk Evaluation of Heavy Metals in the Bottom Sediments: A Case Study of Eutrophic Bukwałd Lake Located in An Agricultural Catchment
Thank you for allowing me to publish my manuscript in the International Journal of Environmental Research and Public Health and for the suggested corrections. Thank you for the reviewer's time and comments. As suggested by the reviewer, the manuscript has been changed (and hopefully improved) in many ways. The responses and changes are highlighted in red.
The answers to specific concerns are included hereafter. (Q: Question; A: Answer)
Reviewer #2
Q1) Please carefully check the spelling and grammar through the whole manuscript. Some examples of typos and spelling errors: L12, L24-25, L31, L49, L65, L67, L78, L95, L99, L110, L117, L311, L405.
A1) Thanks very much for this suggestion. Spelling and grammar of the manuscript have been checked.
Q2) Abstract: The PLI is not presented in the first part of abstract, where the methodology is presented but appears in the last sentence where the results are presented. Rephrase sentence in L16-17. Rephrase “moderate risk of ecological threat”. Rephrase line 24-26.
A2) Many thanks for this suggestion. As suggested by the reviewer, the sentence "To compare the average content of metals in the bottom sediments of lakes, the pollutant loading index (PLI) was used" was added to the abstract section. In accordance with the reviewer's suggestion, the sentence was reworded in lines 16-17 and 24-26.
L16-17 “The calculated RI value was 153.3, which represents a moderate risk of ecological threat”.
L24-26 “The highest CF values were determined for Cr, Ni and Pb and classified as significant contamination. The remaining elements were classified as moderately contaminated.”
Q 3) Add “lake sediments” instead of “bottom sediments” to keywords
A3) Thanks very much for this suggestion. According to reviewer’s suggestion lake sediments” instead of “bottom sediments” to keywords have been added.
Q4) The authors use the term heavy metal, heavy metal pollutant, trace elements, elements. Please use a single term through the whole manuscript. I would suggest trace metals or metals.
A4) Many thanks for this suggestion. As suggested by the reviewer, the term heavy metal, heavy metal pollutant, trace elements has been replaced by trace elements in the manuscript.
Q5) Rephrase L34-35
A5) In accordance with the reviewer's suggestion, the sentence was reworded in lines 34-35. “For this reason, the ecosystem of lakes in areas with a humid tropical climate functions completely differently than in areas with a subtropical climate or in lakes of the temperate zone.”
Q6) L52-53 add relevant reference for the statement
A6) Thanks very much for this suggestion. According to reviewer’s suggestion at Lines 52-53 reference have been added.
They can also have negative impacts on aquatic ecosystems, the food chain, and human health [14].
- Ahamad, M.I.; Song, J.; Sun, H.; Wang, X.; Mehmood, M.S.; Sajid, M.; Khan, A.J. Contamination level, ecological risk, and source identification of heavy metals in the hyporheic zone of the Weihe River, China. Int. J. Environ. Res. Public Health 2020, 17, 1070.
Q7) L53-55 add relevant reference for the statement
A7) AThanks very much for this suggestion. According to reviewer’s suggestion at Lines 53-55 reference have been added.
Numerous studies indicate that only a portion of the heavy metal remains in the aquatic environment, while the majority accumulates in the bottom sediments [15].
Malvandi, H. Preliminary evaluation of heavy metal contamination in the Zarrin-Gol River sediments, Iran. Mar. Pollut. Bull. 2017, 117, 547–553.
Q6) L65 clarify “soil sediment” is soil or sediment?
A7) Thank you very much for this comment. In line 65 it should read bottom sediment, not soil sediment.
Q8) Addition of supplementary parameters such as pH, N, P, C, major elements could give additional information on the sources of metals in the lake sediments
A8) Many thanks for this suggestion. As suggested by the reviewer, the sentence "The pH of the bottom sediments of Lake Bukwałd was neutral and their Ph was at a relatively uniform level and did not exceed 7.40 (Table 7). However, considerable fluctuations were observed in the nitrogen concentrations. Its content in the sediments ranged from 0.30 to 9.30 mg·kg−1. Higher nitrogen concentrations were found in the sediments from the agricultural catchment and the lowest in the sediments collected near the outflow of the lake. The calculated coefficient of variation was high (88%), which means that the obtained results of spatial nitrogen concentration in sediments were characterised by high variability. Such values may indicate an anthropogenic source of nitrogen enrichment in the sediments.” was added.
Table 7. Content N, P, C and pH in the sediments of Lake Bukwałd (g·kg−1 of dry matter).
Heavy Metal |
|
Min |
Max |
Me |
CV |
pH |
- |
7.17 |
7.40 |
7.24 |
- |
N |
3.22±2.83 |
0.30 |
9.30 |
1.73 |
0.88 |
P |
0.71±0.16 |
0.49 |
1.11 |
0.69 |
0.22 |
C |
18.39±2.03 |
15.37 |
22.75 |
17.91 |
0.11 |
Explanations: – arithmetic mean, SD – standard deviation, min – minimum, max – maximum, Me – median, CV – coefficient of variation [%].
Q9) L88-23.5 m does not represent a considerable difference in elevation
A9) Thank you very much for this comment. As the reviewer suggests, 23.5m does not represent a significant height difference. Correct the statement to "The catchment area of the lake is a hilly area with a difference between the highest and lowest point in the catchment area of 23.5 m."
Q10) L97 define “wasteland”
AQ10) Thank you very much for this comment. The name of the use of the catchment area "Wasteland" has been changed to "Fallow".
Q11) L113-115. After drying how were the sediments prepared for analysis? (Crushing? Sieving?).
A11) Thank you very much for this comment. After the samples were dried at room temperature, they were crushed and then their particle size was measured by wet laser diffraction using a Mastersizer 3000 particle analyser.
Q12) The measurement method of organic matter is not presented in the Mat &Met section, yet appear in results. Introduce details on quality control approach used.
A11) Thank you for this suggestion. As per the reviewer's suggestion, the sentence "The organic matter content was quantified on the basis of mass loss after ignition at 550°C" was added to Materials and Methods. The amount of carbon was determined on the basis of the organic matter content with an average factor of 1.724 as described by Pribyl [21]" was added
Q13) L117-sediment sample instead of soil sample
A13) Thank you very much for this comment. In line117 the soil sample was replaced by a sediment sample
Q14) Table 2-delete Scope of
A14) Thank you very much for this comment. In Table 2, "scope of" has been removed.
Q15) Table 3 introduce PLI levels in increasing order, similar to CF and Igeo. Revise and add references for the PLI levels.
A15) Thank you very much for this comment. Table 3 has been corrected and references have been added.
Q16) Table 4. 1 decimal should be sufficient. The table could be presented in a more concise way or as a graph.
A16) Thank you very much for this comment. Table 4 has been corrected
Q17) Table 5. Indicate for each parameter samples that have statistically different parameters (e.g. Tukey pair comparison)
A17) Thank you very much for this comment. Table 5 has been corrected and the sentence "There were no statistically significant differences between the measurement points in terms of bulk density and sediment hydration. Statistically significant differences were found in the mineral and organic matter content of the sediments collected on the west side of the lake, while the bottom sediments at the other points showed no statistically significant differences." was added
Q18) L194-196 please revise or delete, as the comparable values of average and median does not show that metals have comparable concentration in the samples but that the values have a homogenous distribution.
A18) Thank you very much for this comment. Line 194-196 has been removed.
Q19) The low (<1) eigenvalues given by the PCA indicate that this method is not appropriate for your data, or that no significant differences exist. Please revise and compare to other studies in the literature
A19) In order to identify the environmental factors influencing the content of trace metals in the bottom sediments of the studied lake, principal component analysis (PCA) was used, i.e., multivariate statistical analysis using the software Canoco 5.0 for this purpose. The explanation of the variance is 81.16%, which, according to ter Braak, C. J. F., & Smilauer, P. (2002), proves the very good quality of the model used. The range of the PCA axis was set to -1 to 1.
ter Braak, C. J. F., & Smilauer, P. (2002). CANOCO Reference Manual and CanoDraw for Windows User's Guide: Software for Canonical Community Ordination (version 4.5). (Microcomputer Power). www.canoco.com. https://edepot.wur.nl/405659
Q20) L217-use the chemical symbol for Cd and Cu
A20) Thank you for this suggestion. As per the reviewer's suggestion in line 217 the chemical symbol for Cd and Cu is used.
Q21) Compare the pollution indices with those reported from other lakes with similar or different land use
A21) Thank you for this suggestion. As suggested by the reviewer's, the pollution indicators were compared to the results of other lakes with similar and different catchment use.
Similar index values in water reservoirs fed from agricultural, forested or sparsely urbanised catchments were also found in bottom sediments of lakes located under similar geographical and climatic conditions. However, despite the low values of these indices, the pollutant loading index (PLI) determined in the sediments of Lake Bukwałd was 2.49. This suggests that immediate measures should be taken to reduce pollution and prevent the degradation of the catchment. A comparison of the Igeo indices calculated for the sediments of Lake Buchwald with the indices calculated for lake sediments in urbanised catchments showed that their values are mostly higher (Table 11). Only for Cr and Ni were the index values lower than in the studied lake. A similar relationship was found for the contamination factor (CF) [49,56,57,58,59,60].
Table 11. Comparison of the indices of Igeo, PLI and CF, which were determined for other lakes with similar and different land use
The method of using the catchment area of lakes |
Index |
Cd |
Pb |
Zn |
Cr |
Ni |
Cu |
Urbanised catchment area |
Igeo |
2.53-2.72 (2.55) |
1,41-3,62 (2,45) |
3,83-7,14 (4,10) |
-1,3-2,71 (1,01) |
0,11-2,76 (0,97) |
1,10-2,5 1(1,27) |
CF |
1.54-4.62 (2.12) |
0,32-3,93 (1,27) |
16,14-28,61 (17,20) |
1,21-2,15 (1,42) |
1,22-3,34 (1,56) |
9,22-17,43 (10,29) |
|
PLI |
0.69-8.95 (1.5) |
||||||
Agricultural catchment area |
Igeo |
0.10-0.42 (0.11) |
0,11-2,34 (0,60) |
0,01-2,39 (0,58) |
0,02-0,32 (0,17) |
0,05-2,46 (0,42) |
0,11-1,66 (0,54) |
CF |
0.20-4.01 (1.92) |
0,26-3,56 (3,29) |
0,40-2,90 (1,84) |
0,10-3,20 (1,22) |
0,30-16,10 (4,53) |
0,80-2,79 (1,28) |
|
PLI |
0.40-4.90 (1.35) |
Explanations: minimum-maximum (arithmetic mean)
Q22) Emphasize which land use have the most negative impact on the lake sediments.
A22) Many thanks for this suggestion. As suggested by the reviewer, the sentence “The use of the lake basin was found to have a decisive influence on the concentration of trace elements in the bottom sediments of the lake. The highest concentrations of trace metals were found in sediments taken from the side of the catchment area used for agriculture. The bottom sediments from the other sub-catchments (forest areas, wetlands, fallow land and sparsely populated areas) had lower levels of the metals studied.” was added.
Q23) Propose possible mitigation measures.
A23) Many thanks for this suggestion. As suggested by the reviewer, the sentence "In order to limit the deterioration of Lake Bukwałd, a number of measures should be taken. An important element of improvement would be the introduction of preventive measures in the catchment area. Preventive principles include adherence to the principles of good agricultural practise, i.e. precise fertilisation and maintenance of the soil in a high culture that meets the needs of the plants. Agricultural activities should be supported by appropriate meliorations, such as drainage water reservoirs or ditches around the lake. An important direction of activities is also the appropriate design of landscape structure, erosion control methods in the development and use of slopes, and the establishment of biogeochemical barriers (e.g. the introduction of biological development of coastal zones).” was added.

Round 2
Reviewer 2 Report
The authors improved considerably they manuscript.